# Key Disease-Related Genes and Immune Cell Infiltration Landscape in Inflammatory Bowel Disease: A Bioinformatics Investigation

**DOI:** 10.3390/ijms25179751

**Published:** 2024-09-09

**Authors:** Kawthar S. Alghamdi, Rahaf H. Kassar, Wesam F. Farrash, Ahmad A. Obaid, Shakir Idris, Alaa Siddig, Afnan M. Shakoori, Sallwa M. Alshehre, Faisal Minshawi, Abdulrahman Mujalli

**Affiliations:** 1Department of Biology, College of Science, University of Hafr Al Batin, Hafar Al-Batin 39511, Saudi Arabia; ksalghamdi@uhb.edu.sa; 2Department of Clinical Laboratory Sciences, Faculty of Applied Medical Sciences, Umm Al-Qura University, Makkah 24381, Saudi Arabia; rahafhkassar@gmail.com (R.H.K.); wffarrash@uqu.edu.sa (W.F.F.); amshakoori@uqu.edu.sa (A.M.S.); smshehre@uqu.edu.sa (S.M.A.); fominshawi@uqu.edu.sa (F.M.); 3Department of Pathology, School of Medical Sciences, Universiti Sains Malaysia, Kubang Kerian 16150, Malaysia

**Keywords:** IBD, Crohn’s disease, ulcerative colitis, transcriptomics, bioinformatics, molecular signature, immune cell infiltration

## Abstract

Inflammatory Bowel Diseases (IBD), which encompass ulcerative colitis (UC) and Crohn’s disease (CD), are characterized by chronic inflammation and tissue damage of the gastrointestinal tract. This study aimed to uncover novel disease-gene signatures, dysregulated pathways, and the immune cell infiltration landscape of inflamed tissues. Eight publicly available transcriptomic datasets, including inflamed and non-inflamed tissues from CD and UC patients were analyzed. Common differentially expressed genes (DEGs) were identified through meta-analysis, revealing 180 DEGs. DEGs were implicated in leukocyte transendothelial migration, PI3K-Akt, chemokine, NOD-like receptors, TNF signaling pathways, and pathways in cancer. Protein–protein interaction network and cluster analysis identified 14 central IBD players, which were validated using eight external datasets. Disease module construction using the NeDRex platform identified nine out of 14 disease-associated genes (*CYBB*, *RAC2*, *GNAI2*, *ITGA4*, *CYBA*, *NCF4*, *CPT1A*, *NCF2*, and *PCK1*). Immune infiltration profile assessment revealed a significantly higher degree of infiltration of neutrophils, activated dendritic cells, plasma cells, mast cells (resting/activated), B cells (memory/naïve), regulatory T cells, and M0 and M1 macrophages in inflamed IBD tissue. Collectively, this study identified the immune infiltration profile and nine disease-associated genes as potential modulators of IBD pathogenesis, offering insights into disease molecular mechanisms, and highlighting potential disease modulators and immune cell dynamics.

## 1. Introduction

Inflammatory bowel diseases (IBD), including Crohn’s Disease (CD) and ulcerative colitis (UC), are persistent inflammatory conditions affecting the gastrointestinal tract [1]. The etiology of the diseases remains uncertain, but it is thought to involve genetic susceptibility, certain environmental factors, and microbial imbalance [1,2].

Gene expression profiling studies have been conducted to clarify the genetic basis and pathogenesis underlying IBD, through which personalized treatment regimens could be developed [3]. Dragasevic and colleagues revealed elevated expression levels of inflammatory cytokines, including *IL17A*, *IL17F*, and *IL23A* genes, in the inflamed mucosa of individuals with IBD. Moreover, correlations were found between mRNA levels of *IL17A*, *IL17F*, and *IL23A* and clinical disease activity in both CD and UC [4]. In addition, dysregulation of various signaling pathways known to disrupt the intestinal barrier integrity and homeostasis and promote inflammatory response has also been reported, including Wnt, Hippo, and Notch signaling pathways and cytokine–cytokine receptor interaction pathways [5,6,7].

Characterization of IBD’s transcriptional profile using DNA microarray and RNA-sequencing (RNA-seq) technology allowed the identification of gene expression patterns that may indicate the degree of inflammation and potential predictors of the IBD subclasses [8]. Recent studies have investigated the difference in the gene expression profile and the molecular immune mechanisms underlying CD [9] and UC [10]. In CD, altered gene expressions were mainly related to immune responses, including leukocyte migration, neutrophil activity, and cytokine signaling pathways, including interleukin-17 and tumor necrosis factor [9]. Similarly, in UC, genes were enriched in pathways related to the inflammatory response, humoral immune response, and PI3K-Akt signaling [10]. Lu et al. investigated the roles of PANoptosis and autophagy-related genes in UC. Five key genes, including *TIMP1*, *TIMP2*, *TIMP3*, *IL6*, and *CCL2*, were linked to wound healing, cell chemotaxis, and MAPK cascade regulation pathways [11]. Moreover, Nowak et al. used peripheral blood samples to profile the circulating transcriptome of IBD patients and identified altered genes related to monocyte/macrophage polarization. They also highlighted key transcription factors involved in cytokine signaling, such as *NFE2*, *SPI1* (*PU.1*), *CEBPB*, and *IRF2*, with potential translational implications [12]. Recent genomic analyses have further highlighted the role of transcription factors such as *STAT1*, *FOS*, and *IRF1* in regulating immune responses in IBD, particularly through their influence on monocytes [13]. These findings highlight the key role of immune responses and signaling pathways in the pathogenesis of IBD.

Although gene expression studies have provided valuable information, there is still a gap in understanding the molecular modulators that drive IBD and their effect on the immune cells infiltrating the colon. This study aims to thoroughly investigate the molecular basis of IBD, identify key regulatory genes and pathways, and distinguish immune cell populations infiltrating the inflamed and non-inflamed IBD tissues.

## 2. Results

### 2.1. Data Preprocessing and Meta-Analysis of Differentially Expressed Genes (DEGs)

We conducted a comprehensive analysis of 8 IBD publicly available tissue transcriptomic datasets, of which four datasets were for CD and four for UC. CD datasets included 285 inflamed vs. 291 non-inflamed cases and UC included 112 inflamed vs. 79 non-inflamed cases (Table 1). Each dataset was individually preprocessed using appropriate R packages and algorithms implemented in R software version 4.1.3 for background correction and quintile normalization. After applying the statistical threshold, adjusted *p* < 0.05 and |log2FC| ≥ 1, gene expression profiles between inflamed and non-inflamed patients were compared to identify DEGs. To visually represent the differential gene expression patterns, volcano plots were generated for each processed dataset to provide an overview of the upregulated and downregulated genes for CD and UC (Figure 1a and Figure 2a), respectively. The top 10 up- and downregulated DEGs are also shown. To explore the transcriptional relationship between CD and UC DEGs to predict common key genes expressed and understand their regulatory functions, firstly, CD and UC datasets were integrated and a meta-analysis was performed to identify IBD-DEGs across CD and UC samples. A total of 6655 and 6702 DEGs were yielded after a meta-analysis, of which 974 (643 up- and 331 downregulated) and 502 (257 up- and 245 downregulated) were dysregulated across CD and UC datasets, respectively (Figure 1b and Figure 2b). Heatmaps of DEGs in CD and UC were also generated, showing the pattern of expression of the top 10 DEGs (Figure 1c and Figure 2c).

### 2.2. Functional and Pathway Enrichment Analysis

Functional enrichment and KEGG pathways analysis for CD and UC DEGs identified by meta-analysis was performed (Figure 3). DEGs in CD were enriched in biological processes related to cell activation, leukocyte activation and migration, and cytokine production (Figure 3a). UC DEGs were associated with biological processes like metabolic processes, potentially indicating alterations in cellular transport and metabolism as well as inflammatory and immune responses, alongside dysregulated responses to lipopolysaccharides (Figure 3b). When investigating the dysregulated pathways, distinct and shared altered mechanisms were identified. For instance, dysregulated KEGG pathways, such as those involved in cancer, leukocyte transendothelial migration, PI3K-Akt, chemokine, and NOD-like receptor signaling were common in both CD and UC, (Figure 3c). In contrast, CD DEGs showed unique involvement in fatty acid degradation and the Th17 pathway, whereas UC DEGs were distinctly enriched in NF-κB and metabolic pathways (Figure 3c).

### 2.3. Protein–Protein Interaction Network (PPI) Construction and IBD-DEG Identification

To identify hub genes shared by both conditions CD and UC, the DEGs in CD and UC were intersected to identify shared genes using the VENN tool, available in FunRich software (http://www.funrich.org/) (accessed on 18 August 2022). A total of 180 DEGs were common, of which 106 were up- and 74 were downregulated (Figure 4a). The TissuEnrich tool (https://tissueenrich.gdcb.iastate.edu/) (accessed on 2 August 2022) was used to identify the tissue origin of identified IBD-DEGs. Analysis revealed that most of these genes are enriched in the small intestine, appendix, colon, duodenum, and rectum (Figure 4b). To explore the interactions among the identified common genes, a PPI network was constructed using the STRING v.11.5 database and visualized using Cytoscape 3.9 software (Figure 4c). In Cytoscape, the network analyzer function calculated topological parameters to identify high-degree IBD-related hub genes. Local parameters such as degree distribution, centrality betweenness, and closeness were considered to filter IBD-DEGs within the PPI network (Appendix A). Among these hub genes, IL1B, TLR4, ITGB2, CD86, CXCL8, SPI1, HCK, HIF1A, and CYBB showed the higher degree values (Figure 4d and Appendix A).

### 2.4. Disease Modules Screening and Identification of Potential Key Disease-Associated Genes

To further prioritize IBD-DEGs and identify novel disease-candidate genes, two clustering methods were used. First, the MCODE plugin was used, followed by biclustering constrained by networks (BiCoNs) in NeDRex within Cytoscape, to select key substructure clusters from the PPI network. Following the default settings, a total of 4 clusters were obtained: Cluster 1 (MCODE score = 13.7) contained 16 genes, Cluster 2 (MCODE score = 5) contained 5 genes, Cluster 3 (MCODE score = 4) contained 3 genes, and Cluster 4 (MCODE score = 3) contained 6 genes. Additionally, BiCoN identified one cluster containing 18 genes (Figure 5a). The identified clusters were then intersected to obtain common key IBD-DEGs, resulting in the identification of 14 IBD-DEGs (Figure 5b). Eight external validation datasets (GSE117993, GSE4183, GSE13367, GSE16879, GSE36807, GSE38713, GSE6731, and GSE59071) (Table 1) were used to verify the expression pattern of identified IBD-DEGs. All validation datasets underwent similar preprocessing as the primary datasets to ensure the robustness of the comparative analysis. The identified genes demonstrated consistent expression patterns across the validation datasets (Figure 5c).

To get more insight into how these IBD-DEGs contribute to the disease pathogenesis and their mechanistic pathway, the DIAMOnD algorithm in the NeDRex platform was utilized. Briefly, The Get Disease Genes function in NeDRex was used to retrieve known disease genes related to IBD (Figure 6a). The 14 Key IBD-DEGs and known disease genes were used as a seed to construct the disease module using the DIAMOnD algorithm (Figure 6b). Nine out of 14 key IBD-DEGs, including *CYBB*, *RAC2*, *GNAI2*, *ITGA4*, *CYBA*, *NCF4*, *CPT1A*, *NCF2*, and *PCK1*, were connected with known disease genes, then the underlying mechanistic pathways were identified (Figure 6c). Interestingly, besides their implication in IBD pathways, the key IBD-DEGs were also found to be enriched in pathways such as TNF signaling, adipocytokine signaling, colorectal cancer, intestinal immune network for IgA production, leukocyte transendothelial migration, C-type lectin receptor signaling, NOD-like receptor signaling, and VEGF signaling (Figure 6c). Human and mouse phenotypic association of IBD-DEGs was performed using the ToppGene web tool (https://toppgene.cchmc.org/) (accessed on 1 November 2022). Key phenotypic associations include dysregulated inflammatory response, abnormal immune cell physiology, impaired oxidative burst, increased susceptibility to infection, and recurrent abscess formation (Appendix A).

### 2.5. Immune Cell Infiltration Landscape

The influx of activated immune cells into the intestinal mucosa disrupts the epithelial barrier integrity, leading to increased intestinal permeability, a pathophysiological hallmark of IBD. To investigate the infiltration pattern and differences in the immune cell subpopulations, the CIBERSORTx tool was used. Our analysis revealed significant differences in immune cell infiltration between inflamed and non-inflamed IBD tissues. Specifically, inflamed IBD tissues exhibited a higher degree of infiltration in 10 immune subpopulations, including, including neutrophils, activated dendritic cells, plasma cells, mast cells (resting/activated), B cells (memory/naïve), regulatory T cells (Tregs), and M0 and M1 macrophages (Figure 7). In contrast, a lower degree of infiltration was found in 8 immune cell subpopulations in inflamed IBD tissue, including monocytes, activated NK cells, resting dendritic cells, M2 macrophages, gamma delta T cells, follicular helper T cells, CD8 T cells, and CD4 memory resting T cells in IBD inflamed tissue (*p* < 0.05). However, no significant changes were observed in resting NK cells, naïve CD4 T cells, CD4 memory activated T cells, and eosinophils between inflamed and non-inflamed IBD tissues (Figure 7).

## 3. Discussion

IBD is characterized as an autoimmune condition where damage to the intestinal epithelial barrier and high mucosal permeability trigger an immune response, leading to tissue damage, pathological alterations, and the clinical symptoms associated with IBD [27,28]. This study aimed to investigate the molecular mechanisms of IBD by analyzing gene expression data from 767 patients. DEGs were identified, followed by biological pathway analysis, the construction of disease-gene co-expression networks, the identification of key IBD-related genes, and the immune infiltration landscape (Table 2).

A total of 974 genes showed differential expression between inflamed and non-inflamed tissues in CD datasets and 502 genes in UC datasets. Among these, 180 DEGs were found to be shared between UC and CD. Enrichment analysis revealed that DEGs in inflamed CD and UC tissues shared common dysregulated pathways, including pathways in cancer, immune cell migration, PI3K-Akt, chemokine, and NOD-like receptor signaling pathways. For instance, in pathways related to cancer, a previous study highlighted that the prolonged inflammation characteristic of IBD triggers activation of the JAK/STAT3 pathway. This pathway, in turn, promotes the proliferation of epithelial cells, ultimately contributing to cancer development [29]. Another shared pathway between CD and UC was immune cell migration, a key process in the formation of the inflammatory infiltrate in IBD, as circulating white blood cells migrate through the endothelial layer into the colon wall [30]. Previous data highlighted that IBD is characterized by the infiltration of inflammatory T-cells into intestinal tissue [31]. These cells regulate the activity of innate cells such as epithelial cells, fibroblasts, and phagocytes, resulting in sustained hyperresponsiveness to microbial triggers [31,32]. Activated T-cells depend on glucose for energy, regulated by PI3K, which relies on signals received by surface receptors to the AKT pathway. In the current study, the AKT pathway was found to be dysregulated, suggesting that targeting the PI3K pathway could be a promising therapeutic strategy for preventing IBD relapse [33].

Following the identification of DEGs in both CD and UC, as well as the shared enriched pathways between both conditions, we proceeded to identify hub genes that may play significant roles in IBD through the PPI network and topological characteristics. We identified several high-interaction genes, including IL1B, TLR4, ITGB2, CD86, and CXCL8. Some of the identified genes are known to play significant roles in IBD. For instance, interleukin 1 beta (IL1B) serves as a crucial mediator of innate immunity and inflammation, contributing to tissue damage in IBD [34], while toll-like receptor 4 (TLR4) plays a crucial role in recognizing microbial components and triggering inflammatory responses. Once microbes are recognized, TLRs become activated and subsequently trigger innate immunity by inducing IκB phosphorylation, which activates NF-κB [35]. Moreover, TLRs play a role in activating adaptive immunity by promoting the proliferation and differentiation of T cells and controlling the maturation of dendritic cells [36]. Overexpression of TLRs leads to downstream signaling activation and the overproduction of inflammatory cytokines such as TNF, IL-6, and IL-1, ultimately contributing to IBD [37].

To prioritize the identified highly connected genes and uncover novel disease-associated genes, two distinct clustering methods were utilized, resulting in the identification of 14 IBD-DEGs. Validation of their expression in independent datasets confirmed that the expression patterns of these 14 genes were consistent with our initial observations. It is important to highlight that most of the genes require further investigation to determine their implication in IBD, as only *ITGA4* has well-established roles in IBD.

To understand how the identified genes contribute to IBD pathogenicity we employed the DIAMOnD algorithm within the NeDRex. We found nine key IBD-DEGs (*CYBB*, *RAC2*, *GNAI2*, *ITGA4*, *CYBA*, *NCF4*, *CPT1A*, *NCF2*, and *PCK1*) were connected with disease modules. RAC2, a small GTPase belonging to the Rho family, plays critical roles in neutrophil functions. Dysregulation in *RAC2* expression increases susceptibility to bacterial infections and impairment of neutrophil functions [38,39]. *RAC2* was consistently overexpressed across all validation datasets, which may correlate with the observed high infiltration of neutrophils [40]. This finding is consistent with the results obtained from the CIBERSORTx tool, where IBD-inflamed samples showed significantly higher neutrophil infiltration.

Previous studies have consistently reported higher infiltration of neutrophils, macrophages, and activated dendritic cells in affected samples [9,10,41]. Additionally, research focusing on inflammatory cells infiltrating the myenteric plexus layer in UC and CD has shown that B cells and monocytes are highly infiltrated in CD colon tissues, whereas UC exhibits higher infiltration of CD8 cells [42]. In the present study, we observed an increase in B cells; however, the monocytes and CD8 T cells showed less infiltration in IBD-inflamed tissue in comparison to non-inflamed tissues. This contradiction can be attributed to the disease stage or anatomical site investigated, as the previous study focused only on a specific part of the colon [42].

Moreover, RAC2 plays a significant role in regulating the transition of macrophages from the M1 to M2 phenotype, a process crucial for tumor progression, angiogenesis, and metastasis. Loss of *RAC2* expression impedes this transition process [43]. *CYBB* is another important gene as it encodes gp91phox, a key component of the phagocyte oxidase enzyme complex. Mutations in *CYBB* impair the phagocytic capability of cells like neutrophils, monocytes, and macrophages, and have been identified in patients with very early-onset IBD (VEO-IBD) [44]. Impaired phagocytosis leads to recurrent bacterial and fungal infections, predisposing individuals to IBD [45]. Similarly, genes encoding components of the NOX2 NADPH oxidase complex, such as *NCF2* and *NCF4*, show dysregulated expression in the present study. Earlier investigations showed that mutations in those genes, including *CYBA*, have been linked to VEO-IBD and ileal CD, suggesting the role of NOX2 NADPH in IBD pathogenesis [46,47,48]. In the present study, we observed a significant increase in neutrophil infiltration, and interestingly IBD susceptibility genes such as *NCF2* and *NCF4*, involved in neutrophil phagocytic activity [49], were found to be among the top dysregulated identified genes.

Additionally, integrin alpha 4 (ITGA4), is a subunit that forms part of the α4β1 and α4β7, lymphocyte homing receptors that are responsible for the migration of lymphocytes to the intestinal mucosa [15]. The migration of lymphocytes to the inflamed gut is a key feature of IBD pathology. Previous reports showed that targeting leukocyte trafficking with anti-integrin therapy (anti-α4β1 and 7) is effective for patients with IBD and is associated with favorable safety profiles [50,51].

Moreover, *PCK1* is identified as a super-enhancer gene in the human sigmoid colon, the expression of this gene is regulated by the transcription factor (*CEBPB*) [52]. Silencing *CEBPB* resulted in the downregulation of *PCK1* and improved the expression of three epithelial barrier proteins (ZO-1, claudin-1, and occludin). This suggests that the involvement of *CEBPB* in regulating the PCK1 pathway could play a role in the development of IBD via the modulation of epithelial barrier integrity [52]. Furthermore, *CEBPB* deficiency has been reported to impair Treg function and reduce lymphocyte infiltration [53,54]. This suggests that the involvement of *CEBPB* in regulating the *PCK1* pathway could play a role in the development of IBD via the modulation of epithelial barrier proteins. Drugs used as treatment for IBD such as olsalazine, tofacitinib, sulfasalazine, and mesalazine have been found to suppress the expression of *PCK1* and *CEBPB* mRNA in vitro [52]. In the present study, *PCK1* shows very high expression across all validation datasets utilized, which may give a clue to the degree of epithelial barrier damage.

Guanine nucleotide-binding protein G(i) subunit alpha-2 (*GNAI2*), is another key gene involved in regulating macrophage polarization and colitis-associated cancer [55,56]. Increased *GNAI2* signaling promotes an M1 macrophage phenotype, while *GNAI2* signaling deficiency promotes an M2 phenotype [55]. Identified as an IBD susceptibility gene [57], *GNAI2* was upregulated in inflamed tissue of IBD patients, suggesting its significant role in the disease’s inflammatory processes. However, the *GNAI2*-deficient mouse model paradoxically develops spontaneous colitis that resembles human IBD. The degree of colitis correlates with genetic background, microbial composition, and cytokine responses in the colon and cecum [58,59]. For instance, *GNAI2*-deficient C57BL/6 mice are relatively resistant to colitis, whereas *GNAI2*-deficient 129 mice develop IBD earlier and with greater frequency and severity [60]. The upregulation of *GNAI2* may explain the increased infiltration of M1 macrophages in inflamed IBD tissues observed in this study, as previously reported [61].

Lastly, carnitine palmitoyl transferase 1a (*CPT1A*), encodes an enzyme that is involved in fatty acid oxidation. *CPT1A*, located on the outer mitochondrial membrane facilitates the initial transport stage of lipid metabolism [62]. Dysregulation of fatty acid metabolism can lead to an imbalance of pro- and anti-inflammatory mediators [63,64], affecting the nature and severity of inflammatory responses in the intestine [65,66]. In the current study, the *CPT1A* is downregulated. Recently, it was reported that the downregulation of *CPT1A* exerts a protective effect in DSS-induced UC partially through suppressing PPARα signaling [67] and may reduce intestinal damage, inflammation, and oxidative stress, making it a potential target for treatment.

It is important to acknowledge the limitations of our study, including its observational nature based on bioinformatics analysis of previously published datasets. Additionally, factors such as complications, gender, and age were not considered due to the lack of clinicopathological characteristics. The CIBERSORT tool used to identify the immune infiltrating landscape relies on limited genetic data, which could be affected by various predisposing factors, including disease phenotype. Moreover, the findings of this study were not experimentally validated. Nevertheless, our study may still offer valuable insights for further research into the potential of the identified immune infiltration profiles, key modulators, and mechanistic pathways in IBD pathogenesis. Future investigations to validate the identified genes using in vitro and in vivo models of IBD are warranted. Moreover, longitudinal studies involving follow-up of patient cohorts may provide further insights into the dynamic changes in the transcriptomic profile throughout the course of the disease.

## 4. Materials and Methods

### 4.1. Datasets Collection

In this study, we searched the Gene Expression Omnibus (GEO) database (https://www.ncbi.nlm.nih.gov/) (accessed: 13 March 2022) for publicly available tissue microarray datasets using keywords related to inflammatory bowel diseases (IBDs), ulcerative colitis (UC), Crohn’s disease (CD), inflamed, paired samples, healthy margin, uninflamed, and non-inflamed. Only datasets that contain transcriptomic data from inflamed and non-inflamed (healthy margin) were selected. We collected eight tissue datasets, 4 datasets for CD, including GSE179285, GSE75214, GSE186582, and GSE83687, and four datasets for UC, including GSE179285, GSE75214, GSE48958, and GSE9452. For validation purposes, GSE117993, GSE4183, GSE13367, GSE16879, GSE36807, GSE38713, GSE6731, and GSE59071 datasets were used. All the details of the gene expression datasets used in the present study are presented in Table 1. The workflow adopted in this study is shown in Figure 8. No approval from the ethics committee or patient consent was necessary.

### 4.2. Gene Expression Datasets Preprocessing and Identification of DEGs

To account for the differences in microarray platforms used in the collected datasets, each dataset was preprocessed and analyzed separately using the robust multiarray average algorithm (RMA) in R package software version 4.1 (http://www.R-project.org/) (accessed on 15 April 2022). The DEGs were identified by comparing the expression profiles of inflamed and non-inflamed conditions. The DEG analysis employed cutoff criteria of (adjusted *p* < 0.05 and |log2FC| ≥ 1). Quality check reports and density plots of probe intensities were evaluated. Outlier samples were excluded from the downstream analysis to focus on the remaining DEGs, which represent known curated gene sets.

### 4.3. Meta-Analysis of DEGs and Tissue Mapping of the Identified DEGs

To identify common DEGs consistently dysregulated in IBD, a meta-analysis of the processed gene expression datasets for each disease using NetworkAnalyst 3.0. Batch effects were conducted and were corrected using the ComBat function and the False Discovery Rate (FDR) was adjusted using the Benjamini–Hochberg correction in Fisher’s method. Common meta-DEGs were obtained by volcano plots, and upset plots were used for visualization. Furthermore, IBD-DEGs across all CD and UC gene expression datasets were identified by FunRich software [68]. Tissue-specific gene enrichment of IBD-DEGs was performed using the TissueEnrich web tool available at (https://tissueenrich.gdcb.iastate.edu/) (Accessed: 2 August 2022) and Human Protein Atlas (HPO) was used to assess gene–tissue distribution and expression [69].

### 4.4. Functional Enrichment Analysis

To predict the biological functions of the DEGs, a functional enrichment analysis of upregulated and downregulated genes was performed. The Database for Annotation Visualization and Integrated Discovery (David 2021) [70] was used to carry the gene ontology (GO) analysis of common genes within 3 categories (biological process, cellular component, and molecular function) and pathway identification. Functional annotations and pathways with an adjusted *p* < 0.05 were considered statistically significant. We used the open-source ggplot2 package in R for data visualization.

### 4.5. Protein–Protein Interaction Network (PPI) and Selection of IBD Key Genes

To build the protein–protein interaction (PPI) networks, DEGs were submitted to the Retrieval of Interacting Genes (STRING) database https://string-db.org/ (Accessed: 18 August 2022), which provides information on known and predicted protein interactions. For visualization and analysis of the PPI networks, Cytoscape software 3.9 was employed. To characterize key genes in the PPI network and identify topological network properties, including the degree of interaction, closeness, and betweenness, the Cytoscape Analyze Network function was used. To maximize the sensitivity of key IBD-DEGs and module selection, key substructure network and disease modules were identified using MCODE and NeDRex plugins in Cytoscape, respectively, as previously described [71,72]. Briefly, the MCODE algorithm and BiCoN plugins in NeDRex were used to extract PPI modules. The identified modules were then intersected to obtain common key IBD-DEGs. The Get Disease Genes function in NeDRex was then used to retrieve known disease genes. Key IBD-DEGs and known disease genes were used as seeds in the disease module detection (DIAMOnD) algorithm in NeDRex to construct disease modules and the underlying mechanistic pathways, respectively. The DIAMOnD algorithm was run with the following parameters: number of iterations = 20, the weight of seeds = 1, return all edges in the result disease module = False.

### 4.6. CIBERSORTx Estimation of Immune Cell Infiltration

To gain insight into the infiltration patterns and differences in immune cell subpopulations in inflamed and non-inflamed samples, the CIBERSORTx tool was used [73]. According to the tutorial on the website, normalized gene expression matrices were extracted and submitted to the CIBERSORTx online portal (https://cibersortx.stanford.edu/) (accessed on 15 November 2022). The analysis was conducted using the LM22 signature, which identified 22 immune cell subsets for 100 permutations, as previously described [74]. Data was batch-corrected in bulk mode and the results were interpreted in absolute mode, thereby reflecting the precise proportion of each immune cell type within the sample mix. CIBERSORTx results with *p* < 0.05 and correlation coefficient > 0.4 were selected. Results visualization was conducted in GraphPad Prism v9.

## 5. Conclusions

In summary, this study analyzed altered gene expression profiles in IBD patients and identified 14 key disease-related genes, including nine potential disease modulators closely linked with known IBD genes and immune-infiltrated cells (*CYBB*, *RAC2*, *GNAI2*, *ITGA4*, *CYBA*, *NCF4*, *CPT1A*, *NCF2*, and *PCK1*). The immune landscape analysis revealed a higher degree of infiltration of neutrophils, activated dendritic cells, plasma cells, mast cells (resting/activated), B cells (memory/naïve), Tregs, and M0 and M1 macrophages, alongside a lower degree of infiltration of monocytes, activated NK cells, resting dendritic cells, M2 macrophages, gamma delta T cells, follicular helper T cells, CD8 T cells, and resting memory CD4 T cells in inflamed IBD tissues. These findings emphasize the pivotal role of the immune response in IBD pathogenesis, shaped by the dynamic interaction between key genes and immune infiltrated cells. Most of the identified key genes present promising areas for further investigation.

## Figures and Tables

**Figure 1 ijms-25-09751-f001:**
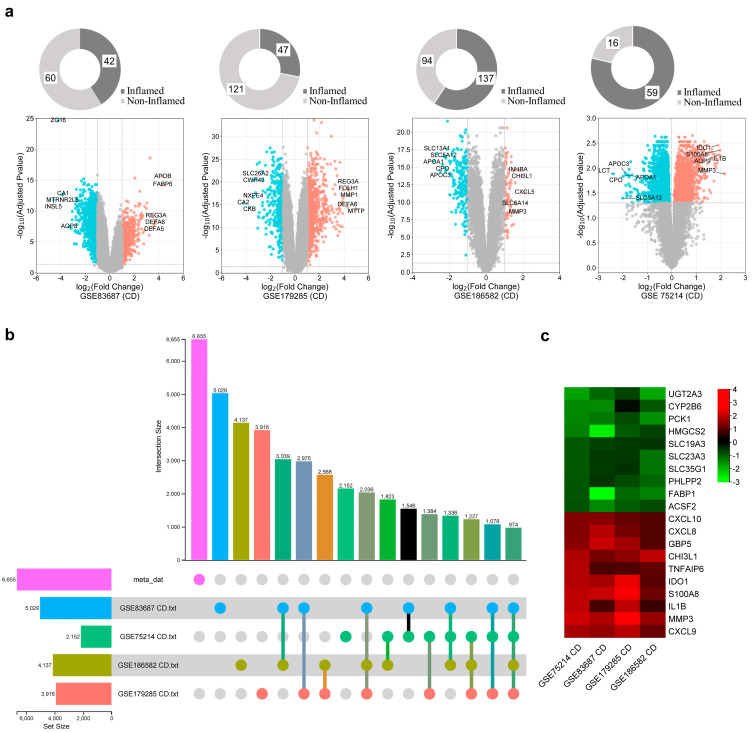
Differential expression analysis of Crohn’s disease (CD). (**a**) Volcano plots across four CD datasets. The colored dots represent the significant DEGs identified at |log2FC| ≥ 1 and an adjusted *p*-value of ≤0.05. The top 10 genes within each dataset are shown. (**b**) The upset plots depict DEG distribution across CD datasets. (**c**) Heatmap of top 10 DEGs identified by meta-analysis. The expression heatmap depicts expression levels of significantly different upregulated and downregulated genes. The color indicates high expression (red) and low expression (green).

**Figure 2 ijms-25-09751-f002:**
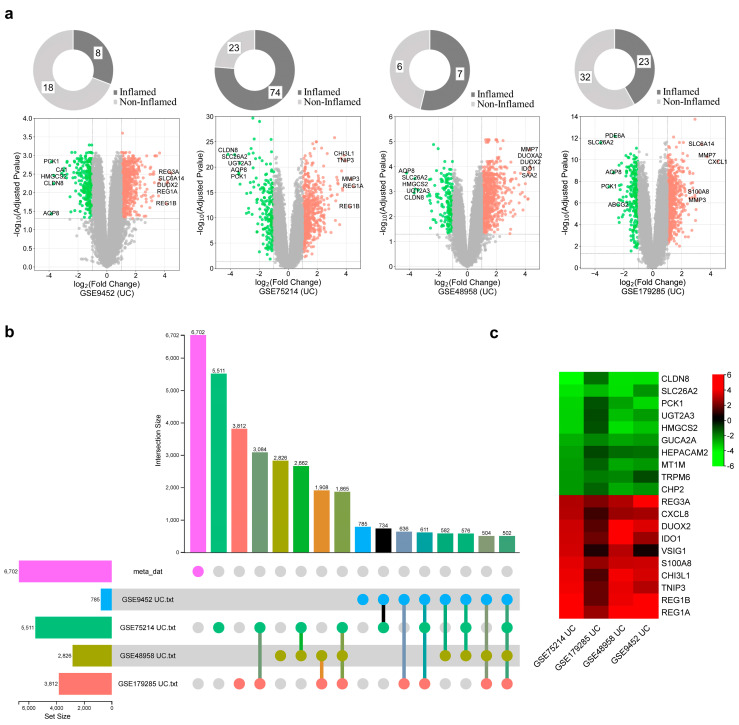
Differential expression analysis of ulcerative colitis (UC). (**a**) Volcano plots across four UC datasets. The colored dots represent the significant DEGs identified at |log2FC| ≥ 1 and an adjusted *p*-value of ≤0.05. The top 10 genes within each dataset are shown. (**b**) The upset plots depict DEG distribution across UC datasets. (**c**) Heatmap of top 10 DEGs identified by meta-analysis. The expression heatmap depicts expression levels of significantly different upregulated and downregulated genes. The color indicates high expression (red) and low expression (green).

**Figure 3 ijms-25-09751-f003:**
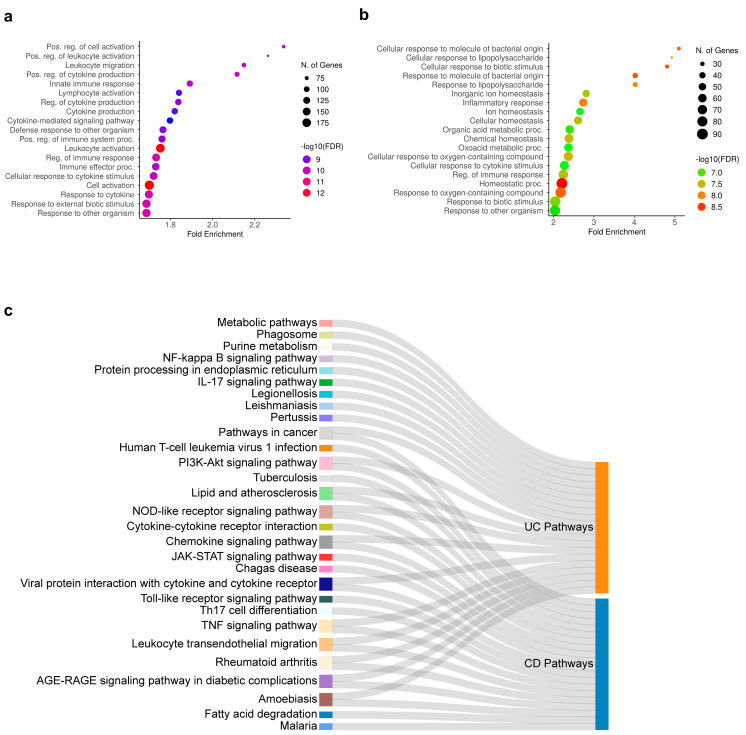
Enrichment analysis of DEGs. Bubble plot of top 20 gene ontology (GO) and KEGG signaling pathways for (**a**) CD and (**b**) UC. The bubble color scaled the enrichment score. The size of the bubbles represents the level of DEG enrichment within each pathway. (**c**) The Sankey plot represents both shared and distinct pathways between CD and UC.

**Figure 4 ijms-25-09751-f004:**
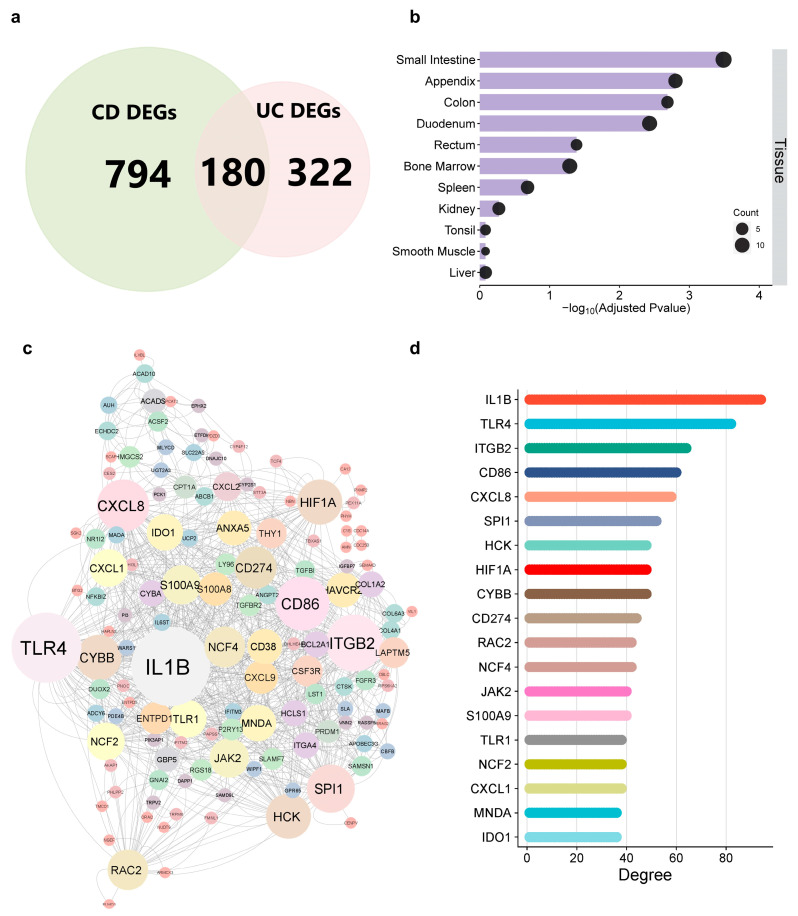
Characterization of IBD-DEGs. (**a**) Venn diagram illustrates shared DEGs between CD and UC, identifying 180 IBD-DEGs with 106 upregulated and 74 downregulated. (**b**) Tissue specificity enrichment analysis of IBD-DEGs shows predominant enrichment in tissues such as the small intestine, appendix, colon, duodenum, and rectum. (**c**) PPI network of IBD-DEGs, displaying top genes with their centrality parameters obtained from network analysis. Node size corresponds to the degree of connectivity. (**d**) Hub genes identified from the PPI network were ranked based on their degree of connectivity.

**Figure 5 ijms-25-09751-f005:**
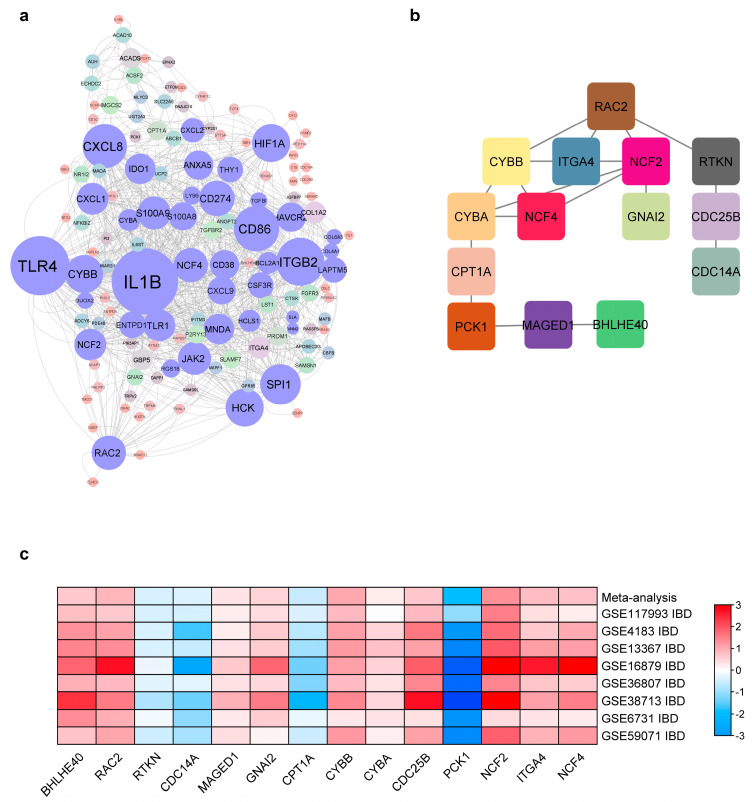
Potential key IB-DEGs. (**a**) PPI network with clusters identified using MCODE and BiCoN clustering methods in Cytoscape. Nodes with lavender color indicate the genes in the clusters and the cluster regions in the PPI network (**b**) 14 key IBD-DEGs identified through the intersection of clusters identified by both methods. (**c**) Heatmap showing the validation of the candidate IBD-DEGs across eight external validation datasets (GSE117993, GSE4183, GSE13367, GSE16879, GSE36807, GSE38713, GSE6731, and GSE59071). Consistent expression patterns of the identified genes were observed across these datasets. Red signifies upregulation, and blue signifies downregulation of expression levels.

**Figure 6 ijms-25-09751-f006:**
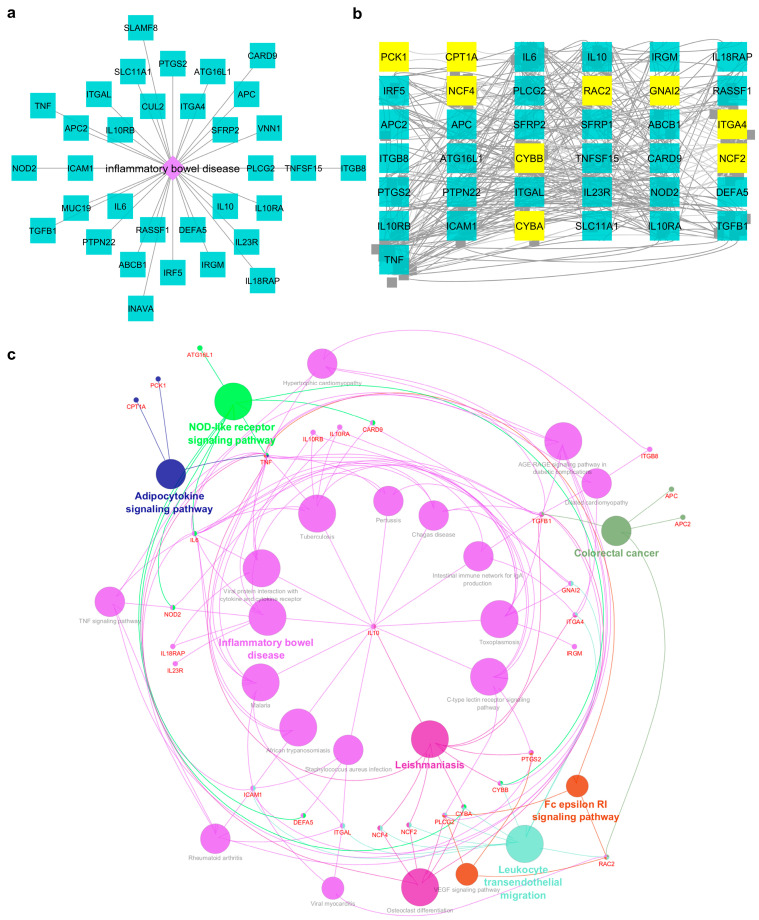
Disease modules of key IBD-DEGs and mechanistic pathways. (**a**) An illustration of IBD-related genes recovered through the Get Disease Genes function in NeDRex platform. (**b**) Disease module obtained from IBD-related genes and potential key IBD genes using DIAMOnD algorithm. Nine out of 14 key IBD-DEGs are present in the subnetwork highlighted in yellow. (**c**) Enriched pathways for key IBD-DEGs.

**Figure 7 ijms-25-09751-f007:**
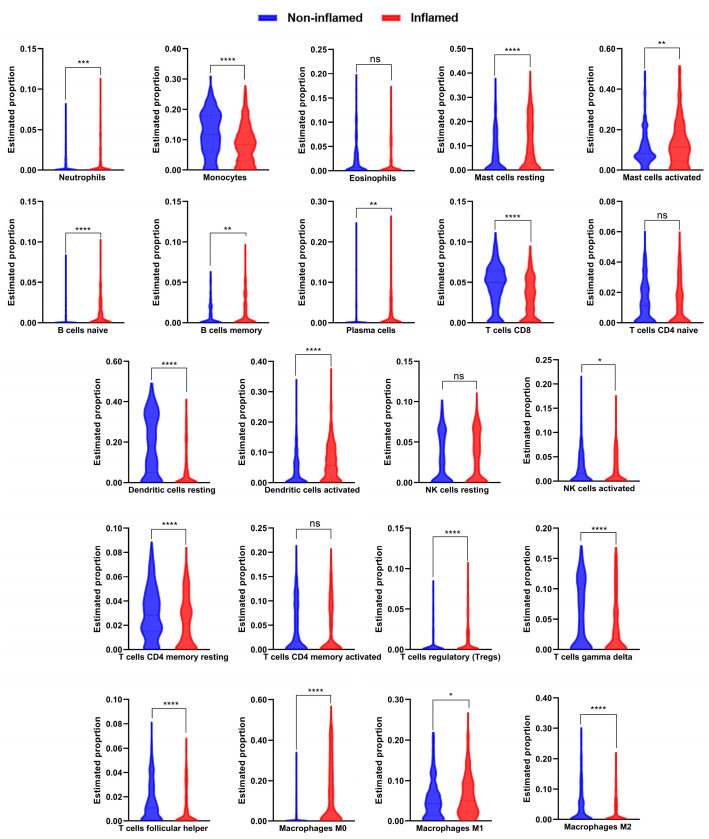
Immune cell infiltration fraction between inflamed and non-inflamed IBD tissues. Violin plots of the proportion of 22 immune cells in inflamed (*n* = 397, red) vs. non-inflamed (*n* = 370, blue) IBD tissues. The red boxplot represents inflamed, and the blue boxplot represents non-inflamed tissues. Significance levels are indicated as follows: ns = non-significant, * *p* < 0.05, ** *p* < 0.01, *** *p* < 0.001, **** *p* <0.0001.

**Figure 8 ijms-25-09751-f008:**
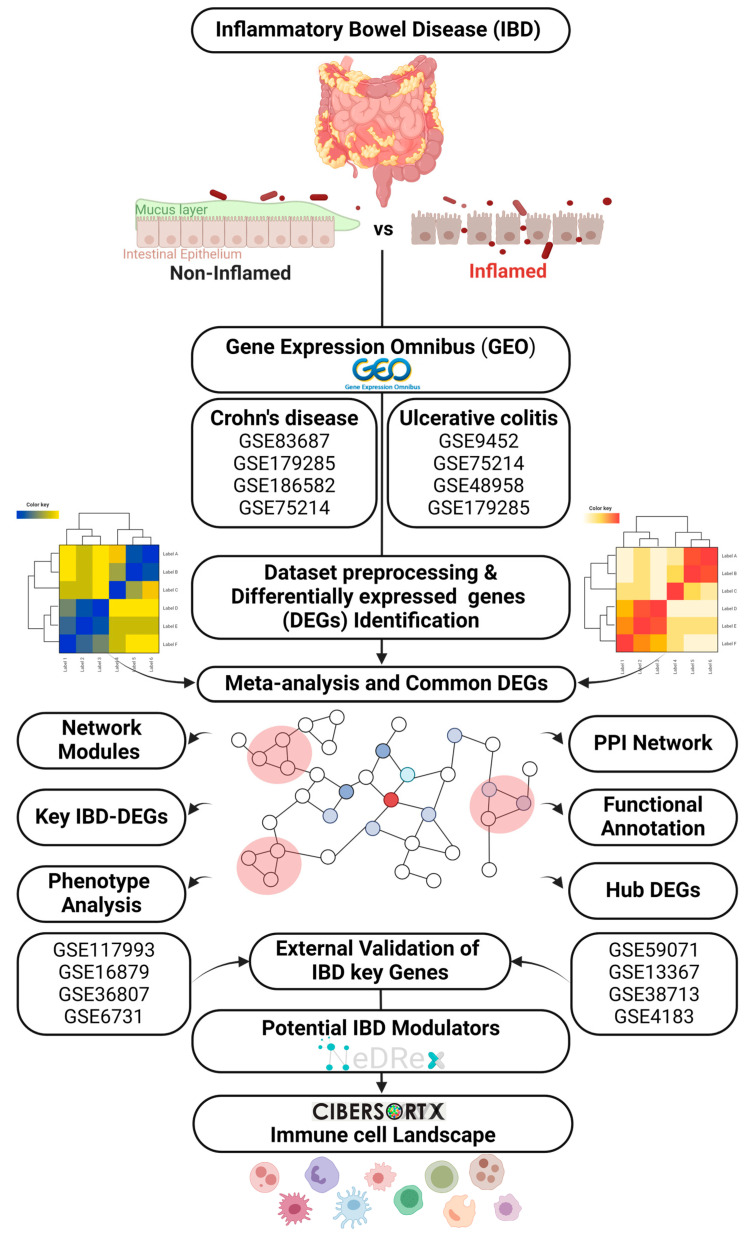
Workflow used in this study.

**Table 1 ijms-25-09751-t001:** Characteristics of transcriptomic datasets included in this study.

**Datasets**					
	**ID**	**Inflamed**	**Non-Inflamed**	**Platform**	**References**
CD	GSE83687	42	60	Illumina HiSeq 2500	[14]
	GSE179285	47	121	Agilent-014850 Whole Human Genome	[15]
	GSE186582	137	94	Affymetrix Human Genome U133 Plus 2.0 Array	[16]
	GSE75214	59	16	Affymetrix Human Gene 1.0 ST Array	[17]
UC	GSE9452	8	18	Affymetrix Human Genome U133 Plus 2.0 Array	[18]
	GSE75214	74	23	Affymetrix Human Gene 1.0 ST Array	[17]
	GSE48958	7	6	Affymetrix Human Gene 1.0 ST Array	[19]
	GSE179285	23	32	Agilent-014850 Whole Human Genome	[15]
	Total	397	370	For Meta-analysis	
**Validation Datasets**					
	**ID**	**Patients**	**Healthy**	**Platform**	
IBD	GSE117993	135	55	Illumina HiSeq 2500 (Homo sapiens)	[20]
	GSE4183	15	8	Affymetrix Human Genome U133 Plus 2.0 Array	[21]
	GSE13367	8	10/9 ^1^	Affymetrix Human Genome U133 Plus 2.0 Array	[22]
	GSE16879	98	12	Affymetrix Human Genome U133 Plus 2.0 Array	[23]
	GSE36807	28	7	Affymetrix Human Genome U133 Plus 2.0 Array	[8]
	GSE38713	23	13	Affymetrix Human Genome U133 Plus 2.0 Array	[24]
	GSE6731	12	17	Affymetrix Human Genome U95 Version 2 Array	[25]
	GSE59071	82	11/23 ^2^	Affymetrix Human Gene 1.0 ST Array	[26]

^1^ GSE13367: 10 healthy samples and 9 inflamed samples; ^2^ GSE59071: 11 healthy samples and 23 inactive samples.

**Table 2 ijms-25-09751-t002:** Summary of key findings on differentially expressed genes, pathways, and immune cell infiltration in inflammatory bowel disease.

Findings	Key Points
DEGs Identified Through Meta-analysis	180 common DEGs were identified between CD and UC
Key Disease-Related Genes	Nine potential IBD modulators (CYBB, RAC2, GNAI2, ITGA4, CYBA, NCF4, CPT1A, NCF2, PCK1)
Pathways Involved	TNF signaling, adipocytokine signaling, colorectal cancer, leukocyte transendothelial migration, NOD-like receptor signaling, and VEGF signaling
Phenotypic Traits	Dysregulated inflammatory response, impaired oxidative burst, increased susceptibility to infection, and recurrent abscess formation
Increased Immune Infiltration	Neutrophils, activated DCs, plasma cells, mast cells (resting/activated), B cells (memory/naïve), Tregs, M0 and M1 macrophages
Reduced Immune Infiltration	Monocytes, activated NK cells, resting DCs, M2 macrophages, T cells (gamma delta, follicular helper, CD8, and CD4 memory resting)
No Change	Resting NK cells, T cells CD4 (naïve/memory activated), and eosinophils

## Data Availability

All datasets analyzed during the current study are publicly available in the GEO repository https://www.ncbi.nlm.nih.gov/geo/ (accessed on 13 March 2022).

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
