# Peer review of "Key Disease-Related Genes and Immune Cell Infiltration Landscape in Inflammatory Bowel Disease: A Bioinformatics Investigation"

_ijms, 2024, doi:10.3390/ijms25179751_

Round 1

Reviewer 1 Report

Comments and Suggestions for Authors

This manuscript is, potentially, a good addition to the scientific literature in relation to the genetics of IBD. However, prior to being accepted for publication, I would suggest that the authors modify the script such that it is more widely acceptable to both a broader and specialized readership, "reads"  better, is scientifically more acceptable and, hopefully, garners a greater number of  citations, when published, than it would in its current state. 

1. The title should be changed; replace "In-Depth" with "A". The reason being that very few scientific studies are "in depth". Faults from-a scientific perspective- can always be found.  Secondly it is up to the reader to decide as to whether or not it is in fact "in-depth" depending on the broad and often intricate knowledge base relating to the study being investigated.

2. There is much repetition, throughout the manuscript in relation to the use of the term "we". Please alter the syntax in order to avoid such repetition!

3. There are many necessary/useful/up-to-date references which are missing. The authors are invited to research the references supplied below and add all of them to both the  main body of the manuscript and, obviously, to the reference section. Moreover, they should cogitate on the idea as to how and if the information in the suggested additional manuscripts can be incorporated in to the text of the current manuscript in order to improve the scientific "substance" of the reported study particularly in relation to the introduction, genes involved (in a more clear/methodical manner) and discussion sections. 

1.       Qin Chen et al., Identification of diagnostic biomarks and immune cell infiltration in ulcerative colitis, Scientific Reports  (2023) 13:6081

2.       Weitao Hu et al., Identification of hub genes and immune infiltration in ulcerative colitis using bioinformatics, Scientific Reports  (2023) 13:6039

3.       Jiali Lu et al. PANoptosis and Autophagy-Related Molecular Signature and Immune Landscape in Ulcerative Colitis: Integrated Analysis and Experimental Validation, Journal of Inflammation Research 2024:17 3225–3245

4.       Jakob J. Wiese et al., Myenteric Plexus Immune Cell Infiltrations and Neurotransmitter Expression in Crohn’s Disease and Ulcerative Colitis, Journal of Crohn's and Colitis, 2024, 18, 121–133

5.       Jasmina El Hadad et al., The Genetics of Inflammatory Bowel Disease, Molecular Diagnosis & Therapy (2024) 28:27–35

6.       Rong Huang et al., Identifying immune cell infiltration and effective diagnostic biomarkers in Crohn’s disease by bioinformatics analysis, Frontiers in Immunology 10.3389/fimmu.14:1162473. doi: 10.3389/fimmu.2023.1162473

7.       Qiuyue Yuan and Zhana Duren, Continuous lifelong learning for modeling of gene regulation from single cell multiome data by leveraging atlas-scale external data, bioRxiv preprint doi: https://doi.org/10.1101/2023.08.01.551575

8.       Xuhong Zhang et al., Bioinformatics Analysis of Immune Cell Infiltration and Diagnostic Biomarkers between Ankylosing Spondylitis and Inflammatory Bowel Disease, Computational and Mathematical Methods in Medicine  2023, Article ID 9065561, https://doi.org/10.1155/2023/9065561

9.       Jan K. Nowak et al., Characterisation of the Circulating Transcriptomic Landscape in Inflammatory Bowel Disease Provides Evidence for Dysregulation of Multiple Transcription Factors Including NFE2, SPI1, CEBPB, and IRF2, Journal of Crohn's and Colitis, 2022, 16, 1255–1268

Comments on the Quality of English Language

Please see comments above. In addition there is no doubt that the grammar/syntax used in the manuscript can be improved.

Reviewer 2 Report

Comments and Suggestions for Authors

In this study, the authors tried to comprehensively investigate the molecular framework that shapes IBD pathogenicity. The authors concluded that they identified the immune infiltration profile and nine disease-associated genes as potential modulators of IBD pathogenesis, offering insights into disease molecular mechanisms, and highlighting potential disease modulators and immune cell dynamics.

Comments:

The reviewer has some concerns as follows:

1.     This study is interesting and has novelty.

2.     In Figure 1, although the author wanted to compare two disease conditions (CD and UC), it seemed confusing, including being difficult to compare, the text font in the figure is too small and unclear, and the legend is unclear. It is recommended to split it into two figures.

3.     The authors provided the information for the characteristics of transcriptomic datasets of this study in the Table S1. However, considering the importance of this information, it is recommended to present it in the main text.

4.     Did the authors conduct gene set enrichment analysis?

5.     Has the authors considered the correlation with immune-related genes?

6.     In Figure 6, what is the number of samples for each analysis?

7.     The references cited in this manuscript are appropriate and relevant to this research.

8.     This manuscript is well-writing, but still needs a revision.

Reviewer 3 Report

Comments and Suggestions for Authors

Dear authors,

Congratulations for the manuscript. However, several edits are required before publication can be considered:

1. In the introduction section you should redo the last paragraph. It should only contain the study hypothesis and objectives.

2. The assessment methods should be moved to the materials and methods sections.

3. I suggest summarizing the key findings in a table, not only the graphics, since some of them are difficult for the general reader.

4. It is necessary to add a paragraph of study limitations in the discussion section.

5. I think the conclusions section does not say much about the main findings, so I suggest redoing it.

Best regards,

Comments on the Quality of English Language

Minor edits required.

Round 2

Reviewer 2 Report

Comments and Suggestions for Authors

This revised manuscript has a great improvement and the reviewer has no further comments.